# ViT-GCT: Enhancing Vision Transformers with a Global Context Token for Face Recognition

## Abstract

Vision Transformers (ViTs) are gaining popularity for a range of tasks beyond image classification, including face recognition (FR). ViTs split an input image into patches and utilize self-attention, enabling interactions among patches to capture both local and global relationships. However, standard ViTs lack strong inductive biases, such as spatial priors, which can make it challenging to efficiently learn both fine-grained local features and coarse global structural patterns, ultimately affecting performance. To address this limitation, we propose to inject global semantic information that provides the model with a holistic signal to guide the learning of spatial relationships. Specifically, we introduce a Global Context Token (GCT) to the ViT architecture for FR. The GCT is a learnable token appended to the input patch sequence and interacts with all patch tokens through self-attention, providing complementary global context and enhancing the discriminative power of the resulting context-aware representations. We empirically proved that ViT with GCT outperforms vanilla ViT for FR on all considered benchmarks. Our analysis of attention maps and patch-wise discriminative ability demonstrates that the GCT directs focus more on the eye regions, which are widely recognized as the most discriminative facial areas for FR, whereas other configurations exhibit a more evenly distributed attention. When compared to previous ViT-based FR works, our approach achieves SOTA results when trained on datasets like MS1MV2 and WebFace4M, ranking first among ViT-based models on the IJB-B and IJB-C benchmarks. These findings highlight the GCT effectiveness in enriching global representation and improving FR robustness.

## 1 Introduction

Vision Transformers (ViT) Dosovitskiy et al. (2021) have recently gained popularity in computer vision for achieving performance comparable to traditional CNNs He et al. (2016), motivating several studies Sun & Tzimiropoulos (2022); Khan et al. (2023); Kim et al. (2024); Dan et al. (2023); Qin et al. (2024); Hosen & Islam (2022); Zhong & Deng (2021); Chettaoui et al. (2025); Islam et al. (2022) to explore their use in FR. Typically, ViT performs image classification by introducing a dedicated class token (CLS) Dosovitskiy et al. (2021); Oquab et al. (2024); Chen et al. (2021), whose final embedding serves as a global image representation. While some works Sun & Tzimiropoulos (2022); Khan et al. (2023); Chettaoui et al. (2025) leveraging ViT for FR rely on the CLS token to learn a compact global descriptor, others Kim et al. (2024); Dan et al. (2023) aggregate all patch embeddings (CPE) to preserve localized cues for discrimination. This divergence highlights a core limitation of standard ViTs, which lack strong inductive biases such as spatial priors Dosovitskiy et al. (2021), making it nontrivial to simultaneously capture fine-grained local features and coarse global structure. Since global and local information can mutually enhance each other, leveraging these complementary cues remains critical to improve discriminative performance Wang et al. (2024); Ebert et al. (2023); Nguyen et al. (2024), motivating our exploration of a unified approach that exploits the strengths of both.

In this work, we propose ViT-GCT, an enhanced ViT token configuration for FR that incorporates an additional Global Context Token (GCT), containing the input image to enrich spatial features with complementary holistic information. We validate our method on multiple FR benchmarks by

comparing it against CLS, CPE, and a novel hybrid CLS + CPE strategy. We analyze how the GCT influences attention maps and enhances the discriminative ability of individual tokens. Additionally, we train our ViT-GCT model on several widely used training datasets to enable fair comparison with prior works. Our results reveal the following key findings:

- Among the ViT token configurations considered, we empirically demonstrate that our method consistently outperforms the others across most benchmarks.
- When trained on datasets such as MS1MV2 and WebFace4M, our model achieves SoTA results, ranking first among ViT-based approaches on the IJB-B and IJB-C benchmarks, thereby demonstrating robustness across different training sets.
- Analysis of attention maps and patch-wise discriminative ability shows that the GCT focuses primarily on the eye regions, which are widely recognized as the most discriminative areas for FR, whereas other configurations distribute attention more evenly.

These results highlight the potential of carefully designed token configurations to enhance global feature representation and suggest promising directions for future research to improve accuracy and robustness in FR.

## 2 RELATED WORK

FR has experienced rapid progress over the last decade, primarily driven by advancements in deep learning and the widespread availability of large-scale annotated datasets Guo et al. (2016); An et al. (2021); Deng et al. (2019a;b); Zhu et al. (2021). Convolutional Neural Networks (CNNs) such as ResNet He et al. (2016), MobileFaceNet Chen et al. (2018), PocketNet Boutros et al. (2022c), and GhostFaceNet Alansari et al. (2023) have dominated the field, achieving SOTA performance on several FR benchmarks Huang et al. (2008); Sengupta et al. (2016); Moschoglou et al. (2017); Zheng et al. (2017); Zheng & Deng (2018); Whitelam et al. (2017); Maze et al. (2018). However, CNNs inherently suffer from limitations in modeling long-range dependencies due to their localized receptive fields. To address this, ViTs, originally proposed for natural image classification by Dosovitskiy et al. Dosovitskiy et al. (2021), have recently emerged as a compelling alternative, offering greater flexibility and the ability to model global contextual relationships, encouraging a range of solutions Sun & Tzimiropoulos (2022); Khan et al. (2023); Kim et al. (2024); Dan et al. (2023); Qin et al. (2024); Hosen & Islam (2022); Zhong & Deng (2021); Chettaoui et al. (2025); Islam et al. (2022) focused on their use in FR.

Typically, ViT performs image classification by introducing a dedicated class token (CLS) Dosovitskiy et al. (2021); Oquab et al. (2024); Chen et al. (2021), whose final embedding serves as a global image representation. Building on this token configuration, several works Sun & Tzimiropoulos (2022); Khan et al. (2023); Chettaoui et al. (2025) have adapted ViT for FR. Sun Sun & Tzimiropoulos (2022) proposed a part-based pipeline where a lightweight CNN predicts facial landmarks, and local patches around them are fed to a ViT for part-aware FR. FRoundation Chettaoui et al. (2025) adapted CLIP Radford et al. (2021) and DINOv2 Oquab et al. (2024) foundation models for FR using Low-Rank Adaptation (LoRA) Hu et al. (2022), demonstrating the potential of leveraging the inherent generalizability of foundation models in low-data availability scenarios. ARTriViT Khan et al. (2023) is a triplet loss-based Siamese network with a ViT as a feature extractor. The Siamese network analyzes a pair of face images as input, extracts the characteristics from these pairs, and uses similarity indexes to evaluate them for FR. While the CLS-based token configuration follows the original ViT setup Dosovitskiy et al. (2021), other approaches Kim et al. (2024); Dan et al. (2023) discard the CLS token and instead aggregate all patch embeddings. To make ViT more resilient to scale, translation, and pose variations, Keypoint-Relative Position Encoding (KP-RPE) Kim et al. (2024) builds on Relative Position Encoding (RPE) in ViTs by introducing Keypoint RPE (KP-RPE), which assigns pixel importance based not only on proximity but also on their positions relative to keypoints, enhancing the model's ability to preserve spatial relationships under affine transformations. TransFace Dan et al. (2023) addresses overfitting and training instability issues often encountered when training ViTs on large-scale datasets such as MS-Celeb-1M Guo et al. (2016) and Glint360KAn et al. (2021). To address these challenges, TransFace introduced Dominant Patch Amplitude Perturbation, which perturbs dominant patches to enhance generalization, and Entropy-based Hard Sample Mining, which weights samples by uncertainty, boosting recognition accuracy

and training efficiency. Given that these works showed that both approaches have their benefits, we propose a novel token configuration that capitalizes on the complementary strengths of each of these concepts, i.e., ViT-GCT.

## 3 METHODOLOGY

This section introduces our proposed method, ViT-GCT, designed to enhance feature representations in ViT-based FR by addressing the limited inductive biases of standard ViTs. In the following, we first provide an overview of the ViT architecture and its FR training pipeline. Then we introduce ViT-GCT, which incorporates global context by using the original input image as an additional patch token, thereby facilitating more effective learning of both local and global patterns.

### 3.1 ViT FOR FR

Given a facial image $X \in \mathbb{R}^{H \times W \times C}$, where $H$, $W$, and $C$ represent the height, width, and number of channels, respectively, the image is divided into a sequence of patches, each of size $P \times P$. These patches are then flattened into vectors, resulting in a patch matrix $\mathbf{x}_p \in \mathbb{R}^{N \times (P^2 C)}$, where $N = \frac{HW}{P^2}$ is the total number of patches. These patch vectors are then linearly projected into a latent embedding space of dimension $D$ using a trainable projection matrix $\mathbf{E} \in \mathbb{R}^{(P^2 C) \times D}$:

$$\mathbf{z}_0^i = \mathbf{x}_p^i \mathbf{E}, \quad \text{for } i = 1, \dots, N. \tag{1}$$

Additionally, a learnable embedding, known as the class (CLS) token Dosovitskiy et al. (2021), is prepended to the sequence of embedded patches, denoted as $\mathbf{z}_0^{\text{CLS}} \in \mathbb{R}^D$. The CLS token serves as a global image representation and is typically used to extract high-level features from the input image Dosovitskiy et al. (2021); Oquab et al. (2024); Chen et al. (2021). To retain positional information, learnable positional embeddings $\mathbf{E}_{\text{pos}} \in \mathbb{R}^{(N+1) \times D}$ are added to the input embeddings $\mathbf{Z}_0$:

$$\mathbf{Z}_0 = [\mathbf{z}_0^{\text{CLS}}; \mathbf{z}_0^1; \dots; \mathbf{z}_0^N], \tag{2}$$

$$\mathbf{Z}_0' = \mathbf{Z}_0 + \mathbf{E}_{\text{pos}}. \tag{3}$$

The resulting sequence of embedding vectors $\mathbf{Z}_0'$, comprising patch embeddings and the additional CLS Token, is then fed into the transformer encoder. The transformer maintains this constant latent vector size $D$ throughout all of its layers. The ViT follows the original transformer encoder design Vaswani et al. (2017), consisting of alternating layers of Multiheaded Self-Attention (MSA) and Multilayer Perceptron (MLP) blocks, each preceded by Layer Normalization (LN) and followed by residual connections. In MSA, we run $k$ heads in parallel, each with its own set of $q$, $k$, and $v$. For an embedding $x$, the computation of the $q$, $k$, and $v$ projection layers in attention head $i$ are:

$$Q_i = \mathbf{x} W_i^Q, \quad K_i = \mathbf{x} W_i^K, \quad V_i = \mathbf{x} W_i^V, \tag{4}$$

where $W_i^Q, W_i^K, W_i^V \in \mathbb{R}^{D \times d}$ are learnable weight matrices, and $d = D/h$ represents the dimensionality of each attention head given $h$ heads. Each head computes scaled dot-product attention:

$$\text{Attention}(Q, K, V) = \text{softmax}\left(\frac{QK^\top}{\sqrt{d}}\right) V. \tag{5}$$

The outputs of all heads are concatenated and projected through a linear transformation to form the final attention output. This mechanism allows the model to jointly attend to information from different representation subspaces, enhancing its ability to capture complex relationships across patches Dosovitskiy et al. (2021). The output is then passed to the MLP, completing the execution of a single transformer block. These repeated attention and MLP operations refine the token representations throughout the encoder depth, ultimately producing contextualized outputs used for downstream tasks Dosovitskiy et al. (2021). In this case, the model processes patch embeddings, generating feature embeddings that capture complex image patterns, with the hidden state of the CLS token serving as the global feature representation of the image.

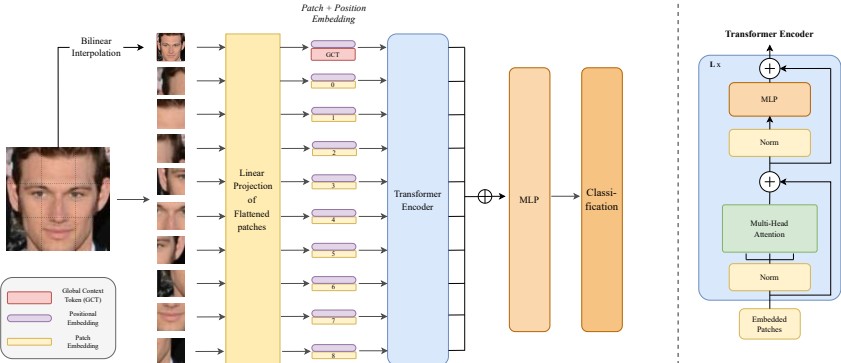

Figure 1: ViT-GCT pipeline. The image is divided into patches, and the GCT is added by resizing the full image to match the patch size. They are projected into the embedding space. Positional embeddings are added to preserve spatial information, and the full sequence is passed through the transformer encoder. The resulting token outputs are concatenated and projected through a linear classification head to produce the final representation. The symbol $\oplus$ denotes the concatenation operator.

To optimize the ViT for FR, following prior works Kim et al. (2024); Chettaoui et al. (2025); Sun & Tzimiropoulos (2022), we enhance the model by incorporating a margin-penalty softmax loss Wang et al. (2018b); Deng et al. (2022). This is achieved by adding a multi-class classification layer and adjusting the loss function to enforce larger margins between classes, which improves the model's discriminative ability for FR tasks Wang et al. (2018b); Deng et al. (2022); Kim et al. (2022); Deng et al. (2019a).

## 3.2 VIT-GCT

Our approach, ViT-GCT, addresses the limited inductive biases of standard ViTs by introducing a Global Context Token (GCT), which enriches patch-based representations and enables the model to more effectively capture both fine-grained local features and coarse global structures. Rather than relying solely on the $\mathrm{CLS}$ token, which serves as the global feature representation of the image, $\mathbf{z}_L^{\mathrm{CLS}}$ or the concatenation of all patch embeddings $\mathbf{z}_L^1, \ldots, \mathbf{z}_L^N$, ViT-GCT introduces a trainable $\mathbf{z}_L^{\mathrm{GCT}} \in \mathbb{R}^D$ that is appended to the set of input patch embeddings. The GCT is similar to the $\mathrm{CLS}$ token in being trainable, but it differs in its initialization. As illustrated in Figure 1, this token is generated by resizing the original input image to match the patch size using bilinear interpolation. GCT aims to provide a complementary global context to enhance feature representation, enriching the detailed spatial information captured by the patch embeddings. The resulting input sequence $[\mathbf{z}_0^{\mathrm{GCT}}; \mathbf{z}_0^1; \ldots; \mathbf{z}_0^N]$ is enriched with positional embeddings to preserve spatial structure and is passed through a stack of transformer encoder layers. These layers apply self-attention to capture relationships across the full sequence, allowing the GCT to interact with all local patches. Finally, all output tokens are concatenated and passed through a linear projection layer to generate the final feature representation.

**Concatenated Patch Embeddings (CPE)** The standard ViT utilizes the $\mathrm{CLS}$ token for classification as the sole output representation for downstream tasks without relying on the other patch tokens. Building on this, a recent work using ViT for FR Kim et al. (2024) has explored an alternative strategy that makes direct use of the patch embeddings. This approach discards the $\mathrm{CLS}$ token and instead leverages all output patch embeddings

$$\mathbf{Z}_L = [\mathbf{z}_L^1; \ldots; \mathbf{z}_L^N] \in \mathbb{R}^{N \times D}, \tag{6}$$

produced by the transformer. This method concatenates all patch embeddings into a single vector of dimension $N \cdot D$, which is then passed through a projection layer to obtain a final feature representation of dimension $D$. The resulting vector serves as the final image representation. As demonstrated experimentally in Section 5, this approach outperforms the use of the $\mathrm{CLS}$ token alone.

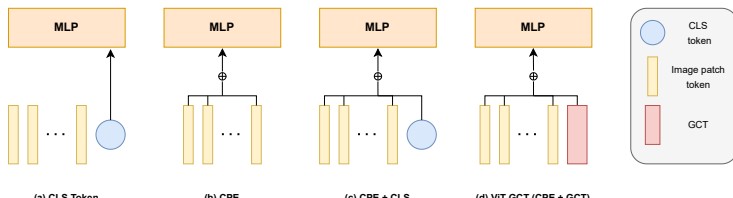

Figure 2: Considered ViT token configurations (a) `CLS` Token: The feature is derived solely from the `CLS` token output. (b) Patch Embeddings: All patch embeddings are concatenated and passed through an MLP. (c) `CLS` + Patch Embeddings: The `CLS` token and patch embeddings are projected and combined. (d) Our ViT-GCT: Concatenates all patch embeddings with a GCT for complementary global information. MLPs adjust input dimensions based on patch count but output a fixed dimension $D$. $\oplus$ denotes concatenation.

**CPE + `CLS`: Hybrid Embedding Strategy**  We explore a hybrid strategy combining CPE and CLS to assess whether their integration yields more effective feature representations. We concatenate all the patch embeddings $\mathbf{z}_L^1, \ldots, \mathbf{z}_L^N \in \mathbb{R}^D$ with the `CLS` token embedding $\mathbf{z}_L^{\text{CLS}} \in \mathbb{R}^D$, resulting in a combined representation

$$\mathbf{z}_{\text{concat}} = [\mathbf{z}_L^{\text{CLS}}; \mathbf{z}_L^1; \ldots; \mathbf{z}_L^N] \in \mathbb{R}^{(N+1)\cdot D}. \tag{7}$$

Although, besides the concatenation, the implementation of the CLS token is the same, it is important to note that, by definition, the CLS token is intended to serve as the output global feature representation of the image. This property is learned during training when the CLS token is used as the model's output. In the hybrid strategy, however, the CLS token is concatenated with all other patches and therefore loses its global representation role, as shown in the supplementary material.

**ViT-GCT**  Since the CLS token loses its role as the global representation in the hybrid strategy, we explore adding a dedicated token initialized with global information from the input image. Based on these insights, we propose ViT-GCT, an enhanced architecture that incorporates CPE alongside an additional token analogous to that employed in the previous hybrid strategy, yet distinct in its design. Unlike the CLS token, this token is generated by resizing the original input image to match the patch dimensions via bilinear interpolation. This token provides global information to the local patch features via self-attention, allowing the model to integrate both local and global cues effectively. In this approach, the trainable GCT embedding $\mathbf{z}_L^{\text{GCT}} \in \mathbb{R}^D$ is concatenated with the patch embeddings. The concatenated sequence

$$[\mathbf{z}_L^{\text{GCT}}; \mathbf{z}_L^1; \ldots; \mathbf{z}_L^N] \in \mathbb{R}^{(N+1)\cdot D}, \tag{8}$$

is then projected into a single feature vector. Figure 2 illustrates the various ViT token configurations we explored, highlighting the progression toward our proposed ViT-GCT approach.

## 4  EXPERIMENTAL SETUPS

**Evaluation datasets**  We evaluate the performance of our proposed ViT-GCT approach on several widely used FR benchmarks. These include Labeled Faces in the Wild (LFW) Huang et al. (2008), Celebrities in Frontal-Profile in the Wild (CFP-FP) Sengupta et al. (2016), AgeDB30 Moschoglou et al. (2017), Cross-age LFW (CA-LFW) Zheng et al. (2017), and CrossPose LFW (CP-LFW) Zheng & Deng (2018). We report verification accuracies (%) following the official evaluation protocols for each of these benchmarks. In addition, we evaluated on large-scale evaluation benchmarks, IARPA Janus Benchmark-B (IJB-B) Whitelam et al. (2017), and IARPA Janus Benchmark-C (IJB-C) Maze et al. (2018). For IJB-C and IJB-B, we used the official 1:1 mixed verification protocol and reported the verification performance as true acceptance rates (TAR) at false acceptance rates (FAR) of $1e-4$ and $1e-5$. These benchmarks were selected because they are commonly used to evaluate the latest advancements in FR and offer a diverse range of use cases Deng et al. (2022);

Wang et al. (2018b); Boutros et al. (2022b); Dan et al. (2023). We also evaluate our model on the more challenging TinyFace Cheng et al. (2018) benchmark, which consists of unconstrained, low-resolution face images. Through this evaluation, we assess the model's robustness on a low-quality face dataset, highlighting its ability to generalize beyond controlled scenarios.

**Training datasets** To train our ViT model, we employ the MS1MV2 Deng et al. (2019a), MS1MV3 Deng et al. (2019b), and WebFace4M Zhu et al. (2021) datasets. These datasets are widely used in FR research, offering a large and diverse set of identities and images. We use them to train our model to enable a wider comparison to SOTA approaches. MS1MV2 is a refined version of the MS-Celeb-1M dataset Guo et al. (2016), curated and refined by Deng et al. (2022), and comprises 5.8M images of 85K identities. MS1MV3 is a cleaned and updated iteration of MS-Celeb-1M, containing 5.2M images of 96k identities. WebFace4M is a subset of WebFace260M Zhu et al. (2021), consisting of 200K identities and 4 million images. All datasets are provided with aligned and resized images (112×112), obtained using five facial landmarks predicted by RetinaFace Deng et al. (2020), following Kim et al. (2024).

**Model Architecture** We use the ViT-B architecture with an input image size of 112×112 and a patch size of 9. The model employs an embedding dimension of 512, a depth of 24 transformer layers, and 8 attention heads. To validate our approach, we evaluated the different ViT token configurations, presented in Section 3.2, while keeping the architecture parameters fixed across all experiments. The variations between configurations are based on how the final feature representation is constructed. In `CLS` configuration, only the `CLS` token is used. In the CPE configuration, the `CLS` token is discarded and only the patch embeddings are used. The CPE + `CLS` configuration combines both the `CLS` token and the patch embeddings. Finally, in our proposed ViT-GCT configuration, an additional GCT is introduced and appended to the sequence of patch embeddings. For all CPE-based configurations, all patch embeddings are concatenated and passed through a projection layer that maps the concatenated features to a $D$-dimensional vector, which is then used for classification.

**Training settings** Following Chettaoui et al. (2025), we employ the CosFace loss function Wang et al. (2018b). Optimization is carried out using the AdamW optimizer Loshchilov & Hutter (2019) with a weight decay of 0.05. The model is trained for 40 epochs using a batch size of 1536. We set the initial learning rate to 0.001 and adopt a polynomial learning rate schedule Mishra & Sarawadekar (2019) with a warmup period of 4 epochs. To improve generalization, we apply a set of data augmentation techniques, following Kim et al. (2024), including horizontal flipping, brightness, contrast adjustments, scaling, translation, RandAugment Cubuk et al. (2020), Gaussian blur, cutout, and rotations up to $20°$. Following prior work Deng et al. (2022); Boutros et al. (2022b); Deng et al. (2019a), we monitor model convergence after each epoch using several face verification benchmarks, including LFW Huang et al. (2008), CALFW Zheng et al. (2017), CPLFW Zheng & Deng (2018), CFP-FP Sengupta et al. (2016), and AgeDB-30 Moschoglou et al. (2017).

## 5 RESULTS

### 5.1 EVALUATING THE IMPACT OF THE GCT INTEGRATION IN VIT

To evaluate the impact of our approach on FR performance, we trained the ViT-B model architecture on the MS1MV2 dataset using the different configurations illustrated in Figure 2: CLS, CPE, the hybrid embedding strategy CPE+CLS, and our ViT-GCT, i.e, CPE+GCT. Based on the reported results in Table 1 that presents the verification performance achieved by each configuration across several evaluation benchmarks, we made the following observations:

- Using only the `CLS` token results in the lowest performance, while CPE consistently outperforms it across all benchmarks. On IJB-C, the performance increases from 96.03% to 96.77% at FAR of $10^{-4}$, and from 91.75% to 93.32% at FAR of $10^{-5}$. Similar gains are observed on IJB-B and Tinyface. These results highlight the limitations of the `CLS` token in capturing subtle yet important identity cues across facial regions, indicating that CPE retains richer spatial features that contribute to a more discriminative final representation. This could explain why relying solely on the `CLS` token results in a weaker global representation compared to approaches that leverage the full set of patch embeddings.

Table 1: Verification performances by ViT-B using the different considered approaches on several evaluation benchmarks. On IJB-B and IJB-C, the results are reported as TAR at FAR of 1e-4 and 1e-5. The GCT configuration results in further performance improvements across most benchmarks, achieving the best results compared to all other considered methods.

| Method | LFW | CFP-FP | AgeDB30 | CALFW | CPLFW | Avg. | IJB-B | | IJB-C | | Tinyface | |
|---|---|---|---|---|---|---|---|---|---|---|---|---|
| | | | | | | | $10^{-4}$ | $10^{-5}$ | $10^{-4}$ | $10^{-5}$ | Rank-1 | Rank-5 |
| CLS | 99.77 | 97.67 | 97.83 | 96.05 | 93.50 | 96.96 | 94.75 | 83.90 | 96.03 | 91.75 | 66.33 | 71.16 |
| CPE | 99.82 | 98.94 | 97.78 | 96.00 | 94.27 | 97.36 | 95.53 | 87.72 | 96.77 | 93.32 | 68.37 | 72.77 |
| CPE + CLS | 99.77 | 98.90 | 98.23 | 96.13 | 94.43 | 97.49 | 95.64 | 87.63 | **96.90** | 93.22 | 69.88 | 73.20 |
| CPE + GCT (our) | 99.83 | 98.90 | 98.03 | 96.25 | 94.57 | **97.52** | **95.82** | **88.79** | 96.86 | **94.05** | **70.41** | **73.79** |

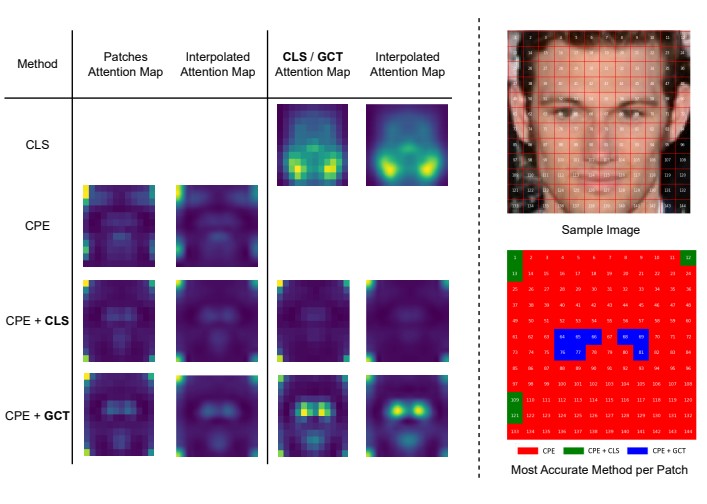

(a) The attention maps of the different approaches      (b) Patches discriminative ability

Figure 3: (a) Illustration of the attention maps of the ViT token configurations. (b) Patch-wise accuracy, each color indicates the configuration with the highest average accuracy.

- The CPE + CLS leads to improvements across several benchmarks when compared to CPE. On IJB-B and IJB-C at FAR of $10^{-4}$, the performance increases from $95.53\%$ to $95.64\%$ and from $96.77\%$ to $96.90\%$, respectively. These results indicate that while CPE captures rich spatial features, the CLS token can provide complementary global context that helps refine the overall representation. On Tinyface, both Rank-1 and Rank-5 accuracies improve from $68.37\%$ to $69.88\%$ and from $72.77\%$ to $73.20\%$, respectively. However, a slight drop is observed at FAR $10^{-5}$ on both IJB-B and IJB-C, suggesting that while the CLS token can contribute useful information, its impact may vary depending on the difficulty and sensitivity of the evaluation setting.

- Using our ViT-GCT results in further performance improvements, achieving the best results across most benchmarks. On IJB-B and IJB-C at FAR of $10^{-5}$, the recognition performance increases from $87.63\%$ to $88.79\%$ and from $93.22\%$ to $94.05\%$, respectively, when compared to CPE + CLS token. At FAR of $10^{-4}$ on IJB-B, the recognition performance increases from $95,64\%$ to $95,82\%$, but we observe a small decrease on IJB-C, where the performance drops from $96.90\%$ to $96.86\%$, which closely matches CPE + CLS. On the Tinyface dataset, there is also a notable improvement, with Rank-1 accuracy rising from $69.88\%$ to $70.41\%$. These results indicate that the GCT is more effective than the CLS token in complementing patch-level features for FR tasks.

We analyze the computational efficiency of ViT-GCT in terms of FLOPs and parameters, highlighting the additional cost introduced by configurations like CPE+GCT due to the inclusion of an extra token, with further details provided in Section 5.4.

## 5.2 QUALITATIVE ANALYSIS OF ATTENTION MAPS: IMPACT OF THE GCT

To better understand the effect of the GCT on training, we visualize the attention maps of the different configurations in Figure 3.(a). These were extracted from the last attention layer and represent the average attention across five benchmarks: LFW Huang et al. (2008), CFP-FP Sengupta et al. (2016), AgeDB30 Moschoglou et al. (2017), CALFW Zheng et al. (2017), and CPLFW Zheng & Deng (2018). The figure shows the average attention received by each patch, as well as the attention from the global token (either CLS or GCT) to all patch tokens, providing insights into how different configurations distribute attention across the image.

First, we observe high attention responses in the corners of the attention maps across all configurations except the CLS-only case. This undesirable effect reduces the interpretability of the attention maps and has been described by Darcet et al. Darcet et al. (2024) as artifacts. These artifacts correspond to high-norm tokens appearing in low-informative background regions of the image, which are repurposed by the model for internal computations. Beyond these artifacts, the attention map of the GCT only in CPE + GCT shows a stronger focus on facial landmarks, whereas other configurations display a more uniformly distributed attention. This localized emphasis influences the attention allocation across all other patches, as visible in their respective patch attention maps, producing a similar overall pattern. Among all the evaluated approaches, ViT-GCT demonstrates the most pronounced attention on the eye region, which is widely recognized as the most discriminative facial area for FR Lederman et al. (2010); Sun & Tzimiropoulos (2022); Wang et al. (2020).

Additionally, Figure 3.(b) shows a color-coded grid illustrating which method achieved the highest average verification accuracy for each patch, based on performance across the same five benchmarks used to generate the attention maps in Figure 3.(a). The grid presents $12 \times 12$ patches extracted from a $112 \times 112$ image using a patch size of 9. Red represents the CPE method, green represents the CPE + CLS method, and Blue represents CPE + GCT (ViT-GCT). ViT-GCT delivers the highest accuracy around facial landmarks, which aligns with the attention map of the GCT presented in Figure 3.(a). Individual patch-wise accuracy maps for each benchmark dataset are provided in Figure 4, highlighting that our approach consistently achieves the highest accuracy around the eye region across datasets.

## 5.3 COMPARISON WITH SOTA ViT-BASED FR

In this section, we compare ViT-GCT with SOTA ViT-based FR methods trained on various datasets. While most of the ViT-based methods adopt the ViT-B architecture, their configurations can differ significantly in terms of attention heads, patch size, embedding dimension, or the depth of the transformer layers, which may influence performance and computational time. For example, our ViT-B backbone has 8 attention heads and a patch size of 9, whereas the KP-RPE Kim et al. (2024) ViT-B has 16 attention heads and a patch size of 8. FRoundation Chettaoui et al. (2025) models are trained with an image size of 224 compared to 112 for the other approaches. We also include ResNet-based FR solutions to provide a broader perspective on the effectiveness of ViT-GCT.

As shown in Table 2, our approach achieves SOTA results when trained on datasets, such as MS1MV2 Deng et al. (2019a) and WebFace4M Zhu et al. (2021). On the IJB-C Maze et al. (2018) benchmark, our approach ranks first among ViT-based methods trained on MS1MV2 Deng et al. (2019a). It also achieves the highest average performance on the five small benchmarks, which are designed to evaluate FR under a wide range of scenarios, such as variations in age and pose, demonstrating the robustness of our method in handling challenging pose variations. Furthermore, our method achieves the best performance on IJB-B and IJB-C among ViT-based methods trained on WebFace4M Zhu et al. (2021). However, it yields comparatively lower performance than ViT-based models trained on the MS1MV3 dataset Deng et al. (2019b). Notably, it performs less effectively on TinyFaceCheng et al. (2018) compared to KP-RPEKim et al. (2024), which leverages strong alignment capabilities that improve performance on low-quality datasets.

## 5.4 COMPUTATION ANALYSIS

In this section, we detail the computational analysis, as stated in Section 5.1, of our proposed ViT-GCT compared to other ViT token configurations, namely CLS, CPE, and CPE+CLS, in terms of FLOPs and the number of parameters. The CLS approach uses a single token of dimension $D$ as the

Table 2: Comparison of the ViT-GCT FR performance in comparison to SOTA ViT-based FR solutions. Based on the different training datasets (as reported by these works), the comparison is made under the dashed line in each section (the best performance is in bold). Results on IJB-B and IJB-C are reported as TAR (@FAR=$1e-4$), other metrics are presented in Sec 4. FR solutions based on non-ViT architectures (ResNet101) are provided to put the ViT performance in scope. ViT-GCT excels when trained on less curated datasets like MS1MV2 and WebFace4M in comparison to MS1MV3. The results mark with * were not reported in the original papers and have been reproduced in the paper using the official models released by the corresponding papers.

| Method | Backbone | Train data | LFW | CFP-FP | AgeDB30 | CALFW | CPLFW | Avg. | IJB-B | IJB-C | Tinyface Rank-1 | Rank-5 |
|---|---|---|---|---|---|---|---|---|---|---|---|---|
| CosFace Wang et al. (2018a) (CVPR 2018) | ResNet101 | MS1MV2 | 99.81 | 98.12 | 98.11 | 95.76 | 92.28 | 96.82 | 94.80 | 96.37 | - | - |
| ArcFace Deng et al. (2019a) (CVPR 2019) | ResNet101 | MS1MV2 | 99.83 | 98.27 | 98.28 | 95.45 | 92.08 | 96.78 | 94.25 | 96.03 | - | - |
| BroadFace Kim et al. (2020) (ECCV 2020) | ResNet101 | MS1MV2 | **99.85** | 98.63 | **98.38** | **96.20** | 93.17 | **97.25** | 94.97 | 96.38 | - | - |
| CurricularFace Huang et al. (2020) (CVPR 2020) | ResNet101 | MS1MV2 | 99.80 | 98.37 | 98.32 | **96.20** | 93.13 | 97.16 | 94.80 | 96.10 | 63.68 | 67.65 |
| URL Shi et al. (2020) (CVPR 2020) | ResNet101 | MS1MV2 | 99.78 | 98.64 | - | - | - | - | - | 96.60 | 63.89 | 68.67 |
| MagFace Meng et al. (2021) (CVPR 2021) | ResNet101 | MS1MV2 | 99.83 | 98.46 | 98.17 | 96.15 | 92.87 | 97.10 | 94.51 | 95.97 | - | - |
| AdaFace Kim et al. (2022) (CVPR 2022) | ResNet101 | MS1MV2 | 99.82 | 98.49 | 98.05 | 96.08 | **93.53** | 97.19 | **95.67** | **96.89** | **68.21** | **71.54** |
| ElasticFace-Cos+ Boutros et al. (2022a) (CVPRW 2022) | ResNet101 | MS1MV2 | 99.80 | **98.73** | 98.28 | 96.18 | 93.23 | - | 95.43 | 96.65 | - | - |
| Face TransformerZhong & Deng (2021) (2021) | ViT-P12S8 | MS1MV2 | 99.80 | 96.77 | 98.05 | 96.18 | 93.08 | 96.77 | - | 96.31 | - | - |
| TransFace Dan et al. (2023) (ICCV 2023) | TransFace-B | MS1MV2 | 99.85* | **99.17*** | **98.53*** | **96.20*** | 92.92* | 97.33* | 95.01* | 96.55 | 64.00* | 68.05* |
| SwinFace Qin et al. (2024) (TCSVT 2024) | Swin-T | MS1MV2 | **99.87** | 98.60 | 98.15 | 96.10 | 93.42 | 97.22 | - | 96.73 | - | - |
| FRoundation Chettaoui et al. (2025) (IMAVIS 2025) | CLIP ViT-B | MS1MV2 | 99.43 | 93.51 | 92.02 | 93.37 | 90.40 | 93.75 | 82.39 | 86.31 | 53.14* | 60.83* |
| ViT-GCT (ours) | ViT-B | MS1MV2 | 99.83 | 98.90 | 98.03 | 96.10 | **94.47** | **97.46** | **95.82** | **96.86** | **70.41** | **73.79** |
| VPL-ArcFace Deng et al. (2021) (CVPR 2021) | ResNet101 | MS1MV3 | 99.83 | 99.11 | **98.60** | 96.12 | 93.45 | 97.42 | 95.56 | 96.76 | - | - |
| AdaFace Kim et al. (2022) (CVPR 2022) | ResNet101 | MS1MV3 | 99.83 | 99.03 | 98.17 | 96.02 | **93.93** | 97.40 | 95.84 | 97.09 | 67.81 | 70.98 |
| Part-based FR Sun & Tzimiropoulos (2022) (BMVC 2022) | Part ViT-B | MS1MV3 | 99.83 | **99.21** | 98.29 | - | - | - | 96.11 | 97.29 | - | - |
| KP-RPE Kim et al. (2024) (CVPR 2024) | ViT-B+KP-RPE | MS1MV3 | - | 99.11 | 97.98 | - | - | - | - | 97.16 | **73.50** | **76.39** |
| ViT-GCT (ours) | ViT-B | MS1MV3 | 99.78 | 99.07 | 98.02 | **96.20** | **94.60** | **97.53** | 95.85 | 97.29 | 69.74 | 73.52 |
| ArcFace Deng et al. (2019a) (CVPR 2019) | ResNet101 | WebFace4M | 99.83 | 99.19 | 97.95 | 96.00 | 94.35 | 97.46 | 95.75 | 97.16 | 71.11 | 74.38 |
| AdaFace Kim et al. (2022) (CVPR 2022) | ResNet101 | WebFace4M | 99.80 | 99.17 | 97.90 | **96.05** | 94.63 | 97.51 | 96.03 | 97.39 | 72.02 | 74.52 |
| KP-RPE Kim et al. (2024) (CVPR 2024) | ViT-B+KP-RPE | WebFace4M | 99.83* | 99.01 | 97.67 | 96.00* | **95.40*** | **97.58*** | 95.49* | 97.13 | **75.80** | **78.49** |
| FRoundation Chettaoui et al. (2025) (IMAVIS 2025) | CLIP ViT-B | WebFace4M | 99.30 | 93.93 | 88.90 | 92.75 | 90.67 | 93.11 | 81.52 | 85.63 | 57.19* | 64.99* |
| ViT-GCT (ours) | ViT-B | WebFace4M | 99.75 | **99.09** | 97.80 | 96.07 | 94.65 | 97.47 | **95.54** | 97.14 | 72.77 | 75.78 |

Table 3: Computation resource comparison in terms of FLOPs and parameter count. GFLOP refers to Giga Floating Operating per Second. We measure it as Rasley et al. (2020).

| Method | GFLOP | # Param | Δ in Param |
|---|---|---|---|
| CLS | 22.98 | 76.08M | ① |
| CPE | 22.89 | 113.83M | ① + 37.75M |
| CPE + CLS | 23.06 | 114.09M | ① + 38.01M |
| CPE + GCT (ours) | 23.06 | 114.09M | ① + 38.01M |

image representation, offering a lightweight design. In contrast, CPE-based methods concatenate all patch embeddings $\mathbf{z}_L^i \in \mathbb{R}^D$ for $i = 1, \ldots, N$, which are then projected to a final feature vector, leading to increased memory and computational load in the final classification stage. Configurations such as CPE+CLS and CPE+GCT (ours) include one additional token, with the CLS token used in the former and the GCT in the latter, resulting in $N + 1$ tokens processed by the Transformer. A detailed comparison of the associated computational costs is presented in Table 3.

## 6 CONCLUSION

In this work, we introduced ViT-GCT, a novel approach that incorporates global semantic information, necessary for holistic understanding, into the ViT architecture for FR, addressing the lack of strong inductive biases, which causes standard ViTs to struggle in jointly capturing fine-grained local details and holistic global structures. The GCT is added to the input patch sequence, where it interacts with all patch tokens via self-attention to deliver complementary context and enrich the overall feature representation. To evaluate our approach, we first evaluated our approach against the traditional CLS token, the CPE method, and a hybrid strategy combining both, demonstrating the superiority of our configuration on most benchmarks. Then, to explore the impact of the GCT, we analyzed attention maps and examined patch-wise discriminative ability, showing that GCT induces a stronger focus on key facial landmarks, whereas other configurations exhibit more uniformly distributed attention. Finally, when evaluated across a broad range of ViT-based FR solutions, ViT-GCT demonstrates superior performance on large-scale benchmarks such as IJB-B and IJB-C, particularly when trained on MS1MV2 and WebFace4M. These results highlight the potential of carefully designed token configurations to enhance global feature representation and suggest promising directions for future research aimed at improving both accuracy and robustness in FR.

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

## A  APPENDIX: IMPACT OF CONCATENATION ON CLS TOKEN AS A GLOBAL FEATURE

As discussed in Section 3.2, the CLS token in standard ViT serves as a learned global feature representation, acquiring this property during training when it is used as the sole output for downstream tasks. The goal of this section is to further examine how this property changes when the CLS token is concatenated with all patch embeddings, as in the hybrid strategy. To this end, we introduce CPE-CLS, a ViT based on the CPE token configuration during training but using only the CLS token as the final feature representation during inference. Table 4 presents the results obtained by both configurations. By comparing the performance of CPE-CLS with that of the standard CLS-based ViT, we show that concatenation disrupts the CLS token's role as a dedicated global descriptor, confirming that its semantic function is compromised when treated merely as an additional patch token.

## B  APPENDIX: IMPACT OF COMBINING CLS AND GCT

To further explore the potential of combining both global tokens, we trained a configuration that retains the CLS token alongside the proposed GCT (CPE+GCT+CLS). As shown in Table 5, the joint use of GCT and CLS produces competitive results and achieves the best performance on IJB-C, although GCT alone remains superior on all other evaluations. This combination comes at the cost of processing one additional token, resulting in a slight increase in computational overhead.

Table 4: Verification performance of ViT-B under the CLS and CPE-CLS across multiple evaluation benchmarks. On IJB-B and IJB-C, results are reported as TAR at FARs of 1e-4 and 1e-5. Concatenating the CLS token with patch embeddings in CPE disrupts its global representation role, leading to degraded performance relative to the CLS configuration.

| Method | LFW | CFP-FP | AgeDB30 | CALFW | CPLFW | Avg. | IJB-B | | IJB-C | | Tinyface | |
| | | | | | | | $10^{-4}$ | $10^{-5}$ | $10^{-4}$ | $10^{-5}$ | Rank-1 | Rank-5 |
|---|---|---|---|---|---|---|---|---|---|---|---|---|
| CLS | 99.77 | 97.67 | 97.83 | 96.05 | 93.50 | **96.96** | 94.75 | 83.90 | 96.03 | 91.75 | 66.33 | 71.16 |
| CPE-CLS | 99,27 | 92,17 | 92,37 | 90,65 | 90,25 | 92,94 | 51.87 | 34.18 | 59.32 | 39.82 | 59.95 | 66.58 |

Table 5: Verification performance of ViT-B under different token configurations, including the combination of GCT and CLS. On IJB-B and IJB-C, results are reported as TAR at FARs of 1e-4 and 1e-5. The joint use of GCT and CLS yields competitive results and achieves the best IJB-C performance, though GCT alone remains superior on all other evaluations.

| Method | LFW | CFP-FP | AgeDB30 | CALFW | CPLFW | Avg. | IJB-B | | IJB-C | | Tinyface | |
| | | | | | | | $10^{-4}$ | $10^{-5}$ | $10^{-4}$ | $10^{-5}$ | Rank-1 | Rank-5 |
|---|---|---|---|---|---|---|---|---|---|---|---|---|
| CPE | 99.82 | 98.94 | 97.78 | 96.00 | 94.27 | 97.36 | 95.53 | 87.72 | 96.77 | 93.32 | 68.37 | 72.77 |
| CPE + CLS | 99.77 | 98.90 | 98.23 | 96.13 | 94.43 | 97.49 | 95.64 | 87.63 | 96.90 | 93.22 | 69.88 | 73.20 |
| CPE + GCT (our) | 99.83 | 98.90 | 98.03 | 96.25 | 94.57 | **97.52** | **95.82** | **88.79** | 96.86 | 94.05 | **70.41** | **73.79** |
| CPE + GCT + CLS | 99.75 | 98.74 | 98.18 | 95.92 | 94.52 | 97.42 | 95.75 | 88.11 | **97.01** | **94.34** | 69.82 | 73.79 |

Table 6: Demographic bias analyses on RFW benchmark as recognition performances (%), STD, and SER across four demographic groups. STD (lower) and SER (closer to 1) values point out that our ViT-GCT results in the lowest bias in comparison to other token setups.

| Method | African | Asian | Caucasian | Indian | Avg. | STD | SER |
|---|---|---|---|---|---|---|---|
| CLS | 98.68 | 98.32 | 99.35 | 98.45 | 98.70 | 0.45 | 2.58 |
| CPE | 99.35 | 98.80 | 99.62 | 99.02 | 99.19 | 0.36 | 3.16 |
| CPE + CLS | **99.47** | 98.72 | **99.68** | **99.08** | **99.23** | 0.42 | 4.00 |
| ViT-GCT (CPE + GCT) | 99.27 | **98.73** | 99.48 | 98.88 | 99.09 | **0.34** | **2.44** |

# C  APPENDIX: BIAS EVALUATIONS

These evaluations are complemented by assessing demographic bias using the Racial Faces in the Wild (RFW) Wang et al. (2019) dataset, reporting verification accuracies and bias evaluation metrics, i.e., the Skewed Error Ratio (SER) and standard deviation (STD). The RFW dataset contains four testing subsets corresponding to the Caucasian, Asian, Indian, and African demographic groups. We present the results as verification accuracies (in %) for each subset, as well as average accuracies. Additionally, we report the standard deviation (STD) and the Skewed Error Ratio (SER), which is defined as $\text{SER} = \frac{\max_g \text{Error}_g}{\min_g \text{Error}_g}$, where $g$ represents the demographic group, as described in Wang et al. (2022); Wang & Deng (2020); Chettaoui et al. (2025). A higher STD value indicates greater bias, and an SER value closer to 1 implies lower bias across groups.

Table 6 shows that ViT-GCT achieves the lowest STD and SER, indicating that it is the least biased model compared to other considered configurations, while still maintaining competitive recognition performance across all demographic groups.

# D  APPENDIX: SUPPLEMENTARY QUALITATIVE ANALYSIS OF ATTENTION MAPS

Section 5.2, presented Figure 3.(b), that showed a color-coded grid illustrating which of the considered token configurations achieved the highest average verification accuracy for each patch, based on performance across five benchmarks: LFWHuang et al. (2008), CFP-FPSengupta et al. (2016), AgeDB30Moschoglou et al. (2017), CALFWZheng et al. (2017), and CPLFWZheng & Deng (2018). This section complements it with Figure 4, which shows individual grids for each benchmark.

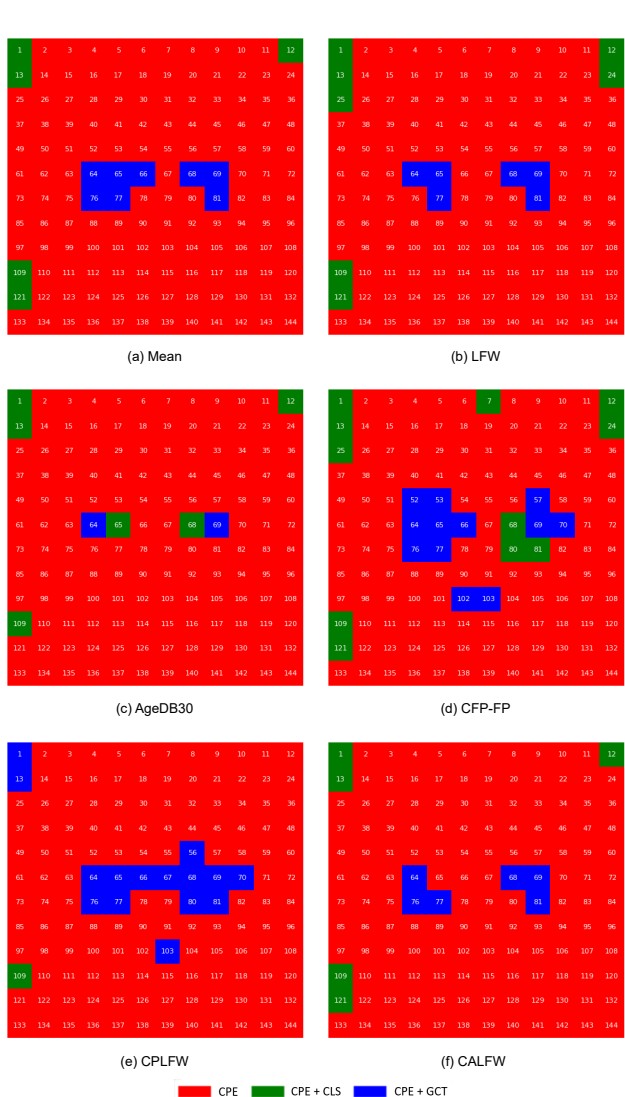

Figure 4: Figure (a) shows the mean patch-wise accuracy map averaged across five benchmark datasets. Figures (b) to (f) display the patch-wise accuracy results for each individual dataset: LFWHuang et al. (2008), CFP-FPSengupta et al. (2016), AgeDB30Moschoglou et al. (2017), CALFWZheng et al. (2017), and CPLFWZheng & Deng (2018). Each patch is color-coded according to the approach that achieved the highest verification accuracy using that patch. Red represents the CPE configuration, green corresponds to the CPE + `CLS` method, and blue represents our CPE + GCT approach. Notably, our approach consistently achieves the highest accuracy around the eye region across datasets.

