# OpenReview forum: "ViT-GCT: Enhancing Vision Transformers with a Global Context Token for Face Recognition"
_ICLR.cc/2026/Conference — Submitted to ICLR 2026_

### Official Review · Reviewer_6zcx · 2025-10-27

**Soundness:** 3
**Presentation:** 3
**Contribution:** 2
**Rating:** 4
**Confidence:** 5

**Summary:**

Vision Transformers (ViTs) have been adapted for face recognition (FR) but struggle to capture both local details and global structures due to a lack of spatial priors. To address this, a Global Context Token (GCT) is introduced as a learnable token that interacts with all patches via self-attention, providing holistic semantic guidance. This improves spatial learning and feature discrimination. Experiments show that ViT with GCT outperforms standard ViTs, with attention focusing more on key regions like the eyes.

**Strengths:**

To introduce inductive bias, a learnable token is appended to the input patch sequence, allowing interaction with all patch tokens through self-attention. This enables the model to effectively identify key facial regions essential for face recognition. This is a good attempt in the field FR.

**Weaknesses:**

Despite the progress of the proposed model, I have several major concerns towards this manuscript:
1. A potential limitation is that the proposed module appears overly general and not specifically designed for face recognition, as it could be applied to other vision tasks without clear specialization or optimization for FR. A more detailed discussion addressing this issue would be valuable.
2. It can also be observed that certain metrics in Table 2 remain sub-optimal, and more detailed explanations are expected to clarify these results.
3. Although the authors have compared their approach with previous methods, a broader review and comparison with additional related works are expected.
4. The avg. performance improvement reported in Table 1 appears marginal, and additional explanations are expected to clarify this observation.
5. Could the authors discuss the potential of the proposed model when applied to synthetic face recognition datasets?

**Questions:**

See above.

---

### Official Review · Reviewer_bgKj · 2025-10-30

**Soundness:** 2
**Presentation:** 3
**Contribution:** 1
**Rating:** 2
**Confidence:** 4

**Summary:**

Authors proposed a learnable Global Context Token (GCT) to the ViT architecture to improve face recognition.  Authors  empirically show that ViT with GCT outperforms vanilla ViT for FR on many benchmarks. Compared to previous ViT-based FR works, their approach achieves good results.

**Strengths:**

1. The paper introduces a Global Context Token (GCT) for Vision Transformers in face recognition, providing a simple yet effective way to enrich global semantic representation.
2. ViT-GCT focuses more on the eye regions, which are widely recognized as the most discriminative facial areas for FR.

**Weaknesses:**

1. The novelty of the idea is quite limited. The concept of global context in ViTs has been around for a while as proposed by [1] given below.

2. The motivation of the proposed approach is almost absent in the introduction section. Authors need to give some insights in the introduction to convince the reader that the choice of GT instead of CLS is a better choice, and also explain which potential benefits it may yield.

3. Is the GT token kept trainable similar to CLS? important explanation is missing. Why is  [CLS]  replaced with [GT]? Why don't authors keep both tokens? The reason to replace CLS with GT is not clear. If GT is trainable, then it is still CLS with a different name?

4. Why does ViT-GT underperform on the TinyFace dataset, despite showing gains elsewhere? A discussion of dataset characteristics or model limitations would strengthen the paper.

5. Lines 285–286 mention that your model operates on 112×112 inputs. What input resolutions and architectures are used in the compared ViT-based baselines? Without this clarification, improvements may be attributed to resolution/model capacity rather than the GT token itself.

6. Lines 298–299 state that the model is trained for 40 epochs. Considering the training datasets (up to 4M–5M images), is 40 epochs sufficient for full convergence? Please provide convergence curves or metrics to support this choice.

7. In Table 2’s caption, you claim that “ViT-GT excels when trained on less curated datasets like MS1MV2 and WebFace4M compared to MS1MV3.” Since MS1MV3 contains 5.2M images, what exactly is meant by less curated in this context?

8. While the difference in performance is negligible in several cases of Table 1, the qualitative results of CPE+CLS are significantly weak. I am unable to fully comprehend this and would appreciate some feedback from the authors. How is the performance almost comparable to ViT-GT when the completely irrelevant patches are demonstrating the highest performance?

Minor Weakness
1. In Figure 3a, the GT token in the CPE+GT configuration exhibits strong attention around facial landmarks. If the GT token alone is this effective, have you evaluated the performance of GT-only classification without additional Tokens?

2. Table 1 shows that CPE+CLS consistently outperforms CLS-only. However, in Figure 3a, the CLS attention map under CPE+CLS looks weaker than CLS alone, suggesting a limited role. Could you clarify this apparent inconsistency?


[1] Hatamizadeh, Ali, et al. "Global context vision transformers." International Conference on Machine Learning. PMLR, 2023.

**Questions:**

Authors should respond to the major and minor weaknesses.

---

### Official Review · Reviewer_NVwW · 2025-11-02

**Soundness:** 2
**Presentation:** 2
**Contribution:** 1
**Rating:** 0
**Confidence:** 5

**Summary:**

This paper presents a novel approach to face recognition by enhancing the Vision Transformer (ViT) architecture through the injection of a global token. This mechanism is designed to improve the model's ability to capture comprehensive global features critical for robust face recognition. The effectiveness of the proposed methodology is empirically validated through extensive experiments performed on large-scale datasets.

**Strengths:**

1.The paper is clearly structured and highly readable, enhancing accessibility for the audience

2.The core idea—injecting a global token into the Vision Transformer (ViT)—is shown to positively impact overall performance

**Weaknesses:**

1.The paper's novelty is marginal. Given that Vision Transformers have been extensively explored and adapted for face recognition, the simple injection of a global token lacks significant technical depth and does not offer substantial new insights to the field

2.The visual clarity and information density are insufficient. Specifically, Figure 1 (the architectural diagram) fails to concisely and effectively convey the paper's key conceptual contribution to the reader.

3.The comparative analysis is incomplete. Figure 3 omits comparisons with several contemporary state-of-the-art methods, such as part-based ViT models and approaches like AdaFace, limiting a fair and comprehensive assessment of the proposed method.

4.The observation concerning attention focus is not a novel finding. The behavioral study of attention maps has been previously examined and documented in related works within the ViT literature

5.The choice of the MS1MV2 dataset limits the demonstrative power of the results. Given its age and widespread use, relying on this dataset alone is insufficient to convincingly showcase the advancement or superiority of the proposed method over previous FR solutions

**Questions:**

please see the weakness

---

### Meta-Review · Area_Chair_PunF · 2026-01-07

**Summary:**

The paper proposes ViT-GCT, adding a Global Context Token (GCT) to a ViT-based face recognition model. Reviewers acknowledge small empirical gains but consistently raise concerns about limited novelty/insight, incomplete/insufficient comparisons to contemporary methods, and multiple clarity/fairness issues in the experimental setup and presentation.

**Reviewer Concerns:**

No rebuttal provided, so key concerns remain outstanding:

1. Limited novelty / technical depth: the contribution is viewed as an incremental token modification without substantial new insight.

2. Comparisons incomplete / not comprehensive: missing or insufficient comparisons to several recent strong baselines and variants.

3. Experimental clarity/fairness: unclear justification of design choices (CLS vs GT), unclear baseline parity (e.g., resolution/capacity), and requests for stronger evidence/analysis (e.g., convergence, dataset-specific behavior such as TinyFace).

4. Interpretation claims: attention-map observations are not considered novel and need stronger support/positioning.

**Reviewer Scores:**

NVwW: stays 0 (strong reject)

bgKj: stays 2 (reject)

6zcx: stays 4 (borderline / below threshold)

---

### Decision · Program_Chairs · 2026-01-26

Reject